# Potential Geographical Distribution of *Lagerstroemia excelsa* under Climate Change

**Siwen Hao** [1,2], **Donglin Zhang** [3] **and Yafeng Wen** [1,2,*]

1 College of Landscape Architecture, Central South University of Forestry and Technology, Changsha 410004, China
2 Hunan Big Data Engineering Center of Natural Protected Areas and Landscape Resources, Changsha 410004, China
3 Department of Horticulture, University of Georgia, Athens, GA 30602, USA; donglin@uga.edu
* Correspondence: wenyafeng7107@163.com

**Abstract:** As a popular ornamental plant and an effective species for controlling rocky desertification, the identification and protection of potential habitats of *Lagerstroemia excelsa* habitats hold significant importance. To gain a comprehensive understanding of the natural resources and growing conditions for *L. excelsa*, predictive modeling was employed to estimate the potential geographical distribution of the species during the Mid-Holocene (MH), the present, and the years 2050 and 2070. The projection was based on current occurrences, and we selected the relevant environmental attributes through the Pearson analysis and the Maximum Entropy Model (MaxEnt). The analysis revealed that temperature and precipitation are the primary environmental factors influencing *L. excelsa* distribution, with the Wuling Mountains identified as a center distribution hub for this species. The anticipated suitable area for *L. excelsa* is expected to experience marginal expansion under future climate scenarios. These results are invaluable for guiding the protection and sustainable utilization of *L. excelsa* in the face of climate change. Additionally, the data generated can be leveraged for enhanced introduction, breeding, selection, and cultivation of *L. excelsa*, taking into account the challenges posed by global warming.

**Keywords:** crape myrtle; global warming; MaxEnt model; potential suitable area; sustainable utilization





## 1. Introduction

Climate change presents significant challenges to both ecosystems and human health [1]. Various weather events, including extreme temperatures, heat waves, floods, storms, and fires, manifest climate change. Plant distributions are significantly affected by these shifts, given their limited migration ability [2]. Consequently, the suitable habitats for many plants are undergoing changes and shifts, placing some species at risk of extinction [3]. Despite these challenges, it is crucial to recognize the pivotal role that plants play in ecosystem services, actively contributing to the carbon and nitrogen cycles and managing runoff water [4]. The alteration of plant habitats has far-reaching consequences on ecosystem stability [5], affecting not only other plants but also animals, thereby posing potential threats to public health [6]. Therefore, gaining a comprehensive understanding of the fitness needs of plants and their potential geographical distribution under the influence of climate change is of paramount importance [2]. The number of publications focusing on plants and their potential geographical distribution is on the rise [7]. However, the majority of existing studies tend to concentrate on rare and endangered plants, with less attention given to widely utilized ornamental species. Crape myrtle (*Lagerstroemia*), as one of the most popular summer ornamental plants, has gained widespread use globally [8,9]. Its cultivation in China dates back to the Tang Dynasty, with crape myrtle being extensively planted throughout the country [10]. Therefore, it is essential to conduct species distribution

modeling for crape myrtle. However, to the best of our knowledge, there is still a lack of distribution modeling for crape myrtle species under climate change.

*Lagerstroemia excelsa* (Dode) Chun ex S.K. Lee & L.F. Lau, native to China, is a towering tree of the *Lythraceae* family with a prominent ecological niche (Figure 1) [11]. It is widely distributed in mountainous or valleys in Sichuan, Guizhou, Hunan, Hubei, and Chongqing. Apart from its considerable ornamental and cultural significance, this species has substantial timber value. This is attributed to its straight and robust trunk, as well as its extended life cycle [12]. Moreover, *L. excelsa* boasts remarkable ecological value, demonstrating robust resistance to harmful gases like $SO_2$, $Cl_2$, $NH_3$, HF, and HCl [13]. This characteristic makes it an exceptional anti-pollution plant suitable for cultivating in factories, mining areas, and regions affected by rocky desertification. Selecting a case study plant for predicting distribution should hinge on its exceptional research value, avoiding widespread use in urban environments and limited discussion in publications. *L. excelsa* stands out as a flagship species with substantial ecological and landscape importance, yet it remains relatively understudied [14]. Therefore, there is a pressing need to investigate the potential distribution of *L. excelsa* under future climate change. Selecting *L. excelsa* as the study subject not only addresses the gap in crape myrtle species distribution modeling but also vividly illustrates the resilient response of resistant plants to climate change. This exploration is not only crucial for the species' utilization and conservation but also holds significance in the broader context of climate change mitigation and the control of rocky desertification.

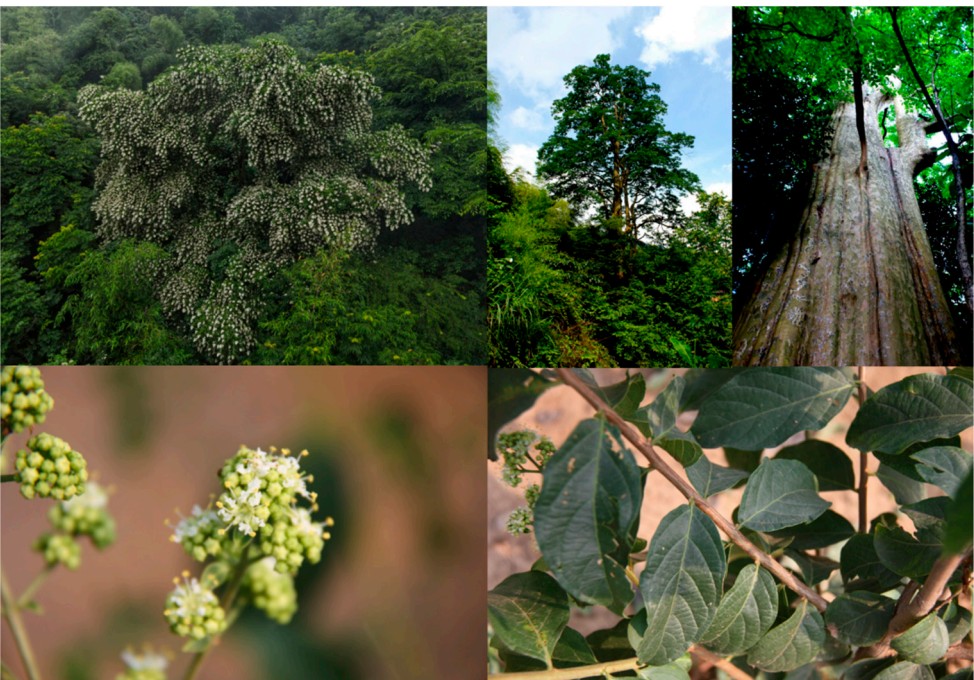

**Figure 1.** The full bloom, habit, stems, foliage, and inflorescence (clockwise) of *Lagerstroemia excelsa*.

Constructing an ecological niche model (ENM) using occurrence data and environmental attributes proves to be an effective approach for delineating potential geographical distributions under varying climates [15,16]. Several species distribution models (SDMs), such as the Bioclimate analysis and prediction system (BioClim) [17,18], genetic algorithm for rule-set production (GARP) [19], and maximum entropy (MaxEnt) [20], are commonly employed in ENM. Among these, MaxEnt stands out due to its advantages of requiring a smaller sample size, offering greater stability, and delivering superior prediction results. Consequently, it has emerged as the most widely utilized SDM [21–24]. Given MaxEnt's demonstrated benefits, we opted for this model to predict the potential geographical distribution of *L. excelsa*.



This study identified the main environmental factors influencing the distribution of *L. excelsa* and predicted the suitable area of *L. excelsa* under different climate scenarios, through the Pearson analysis and the MaxEnt model. The aim of the study is to answer the following three questions: (1) What are the main environmental factors affecting the distribution of *L. excelsa*? (2) What happened and will happen to the suitable area of *L. excelsa* under climate change? (3) What should be the future conservation and utilization for *L. excelsa* under climate change? The study is the first that systematically identifies the potential geographical distribution of *L. excelsa.* It could not only help future researchers better understand the resource distribution of *L. excelsa*, but also provide evidence-based suggestions for better *L. excelsa* introduction, breeding and selection, and cultivation with consideration of global warming.

## 2. Materials and Methods

### 2.1. Species Occurrence Data Collection

Data on the occurrence of *L. excelsa* were gathered through searches on the Global Biodiversity Information Facility (GBIF, http://www.gbif.org (accessed on 5 June 2022)), Chinese Virtual Herbarium (CVH, http://www.cvh.org.cn/ (accessed on 5 June 2022)), Plant Plus of China (PPC, http://www.iplant.cn (accessed on 5 June 2022)), and China National Knowledge Infrastructure (CNKI, https://www.cnki.net (accessed on 5 June 2022)). Additionally, field investigations were conducted to supplement the dataset. To ensure the accuracy of sample point information, Google Earth was utilized for the selection and calibration of coordinates. Duplicate points and points not within the distribution range of *L. excelsa* were removed. During this process, twelve duplicate points, likely arising from specimens of the same plant being collected by different individuals at different times, were excluded. In the end, a total of fifty-six distribution points were compiled from online sources. Due to the lack of recent information in the online collection of site data and the incomplete coverage of the entire distribution range of *L. excelsa*, we conducted field surveys between June and October 2022 and from April to June 2023. In order to comprehensively gather distribution point information for *L. excelsa*, enhancing the accuracy of habitat predictions, an additional 21 points were obtained through these field surveys. This approach aided in the removal of duplicate and inaccurately distributed points. Moreover, to mitigate overfitting in the MaxEnt model resulting from clustering effects [25], we retained only one data point within each 1 km × 1 km grid. After thorough curation, a total of 71 effective geographical distribution points for *L. excelsa* were meticulously selected for further analysis (Figure 2).

### 2.2. Environmental Attributes Collection and Selection

Environmental attributes were chosen based on the habitat characteristics of *L. excelsa*. Environmental attributes were gathered from global climate and weather data sources (WorldClim, http://www.worldclim.org/ (accessed on 23 May 2023)), covering 19 environmental variables (bio1-bio19) and 1 terrain variable (Elevation, Ele) as detailed in Table 1. The Mid-Holocene (MH) data were acquired from the MIROC model because it better represents the climatic characteristics of East Asia. Current climate data were sourced for the period between 1970 and 2000. The selection of General Circulation Model (GCM) predictions under Representative Concentration Pathway (RCP) scenarios was based on previous studies [26]. The BBC_CSM model, developed by the National Climate Center of China, was employed due to its superior capability in simulating the East Asian climate compared to other climate models [27]. To analyze the behavior of the subject, we utilized future climate projections encompassing the 2050s (averaging from 2041 to 2060) and the 2070s (averaging from 2061 to 2080). The scenarios RCP2.6, RCP4.5, and RCP8.5 represent low, moderate, and high carbon emission scenarios, respectively. A total of eight simulated climate scenarios were considered.

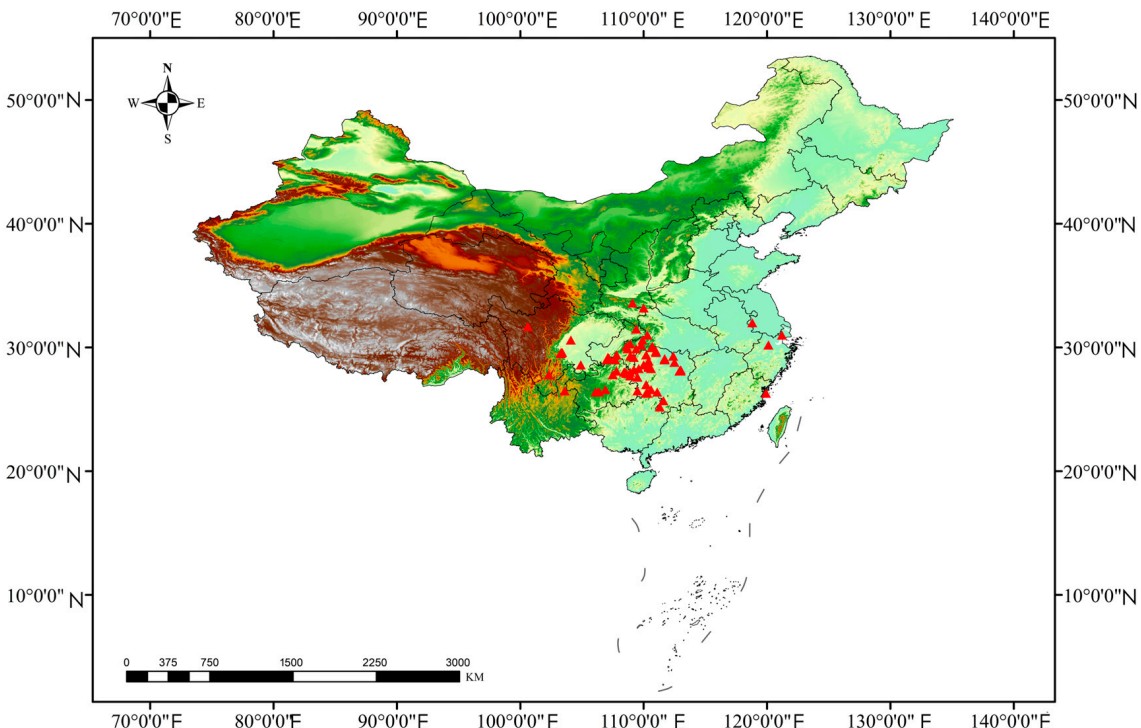

**Figure 2.** Selected distribution points of *Lagerstroemia excelsa* for this study.

**Table 1.** Environmental attributes for geographical distribution of *Lagerstroemia excelsa*.

| Attributes | Summary |
|---|---|
| bio1 | Annual mean temperature |
| bio2 | Mean diurnal range |
| bio3 | Isothermality |
| bio4 | Temperature seasonality |
| bio5 | Max temperature of warmest month |
| bio6 | Min temperature of coldest month |
| bio7 | Temperature annual range |
| bio8 | Mean temperature of wettest quarter |
| bio9 | Mean temperature of driest quarter |
| bio10 | Mean temperature of warmest quarter |
| bio11 | Mean temperature of coldest quarter |
| bio12 | Annual precipitation |
| bio13 | Precipitation of wettest month |
| bio14 | Precipitation of driest month |
| bio15 | Precipitation seasonality coefficient of variation |
| bio16 | Precipitation of wettest quarter |
| bio17 | Precipitation of driest quarter |
| bio18 | Precipitation of warmest quarter |
| bio19 | Precipitation of coldest quarter |
| Ele | Elevation |

Given that all 71 *L. excelsa* occurrence data points were located in China, only China's data were extracted using the mask tool in ArcGIS. To identify the primary environmental factors influencing *L. excelsa* distribution and mitigate model overfitting due to collinearity, Pearson correlation analysis was applied. A Pearson correlation matrix was constructed, wherein all environmental factors underwent pairwise comparisons to analyze their correlations and significance. Following the findings of previous studies, the criterion applied was that if the absolute correlation value between two variables exceeded 0.8, only one of the variables was selected. All environmental attributes were collected at a spatial resolution of 30″ (approximately 1 km).

### 2.3. Potential Geographical Distribution Prediction

MaxEnt 3.4.1 (http://biodiversityinformatics.amnh.org/open_source/maxent/ (accessed on 1 June 2022)) was employed to construct the predictive model, and ArcGIS 10.2.2 (Esri, Redlands, CA, USA) was used to visualize the simulation results and delineate habitat suitability. Furthermore, to mitigate issues of collinearity among environmental attributes, we selectively utilized only those attributes that passed the Pearson correlation analysis ($|r| < 0.8$). To significantly enhance the predictive accuracy of the MaxEnt model, improve model convergence, optimize parameters, and ensure model stability [28], we conducted the operational process ten times, utilizing 75% of the data for training and 25% for verification. Enabling the model to glean insights from the predominant dataset while autonomously validating its performance on unseen data, we optimized the model's parameters during training to bolster its predictive prowess using environmental variables and species occurrence data. The subsequent verification phase gauged the model's adaptability to novel data. Employing diverse evaluation metrics, we fine-tuned the model for optimal performance. The outcomes from each iteration were amalgamated, providing comprehensive insights into the overall resilience and precision of the MaxEnt model in forecasting species distribution. Five analyses were conducted to predict the potential geographical distribution of *L. excelsa*: (1) Receiver Operating Characteristic (ROC): This metric was utilized to assess the accuracy of the simulation results. (2) Jackknife Test: This test calculated the contribution rate and replacement importance value of each environmental attribute, evaluating their impact on the geographical distribution of *L. excelsa*. (3) Jenks Natural Breaks: This method was employed as an effective tool for classifying suitable areas. (4) Change in Distribution Area: This analysis reflected alterations in the distribution of suitable areas across different periods. (5) These analyses collectively provided a comprehensive understanding of *L. excelsa*'s potential geographical distribution under varying climatic conditions.

### 2.4. Research Framework

The research was structured into two main components: the Pearson correlation analysis and MaxEnt modeling. The research framework is outlined in Figure 3. The process began with the collection of *L. excelsa* occurrence data and environmental attributes. Subsequently, Pearson correlation analysis was conducted using the R to select correlated attributes. Utilizing these identified attributes, we constructed a MaxEnt model to predict the potential geographical distribution of *L. excelsa* in the Mid-Holocene (MH), present, and future periods. Finally, the prediction results were visualized. The entire analysis was executed using R 4.3.1 (R Core Team, Vienna, Austria, 2019) and MaxEnt, and the visualizations were created using ArcGIS 10.8 (Esri, Redlands, CA, USA).

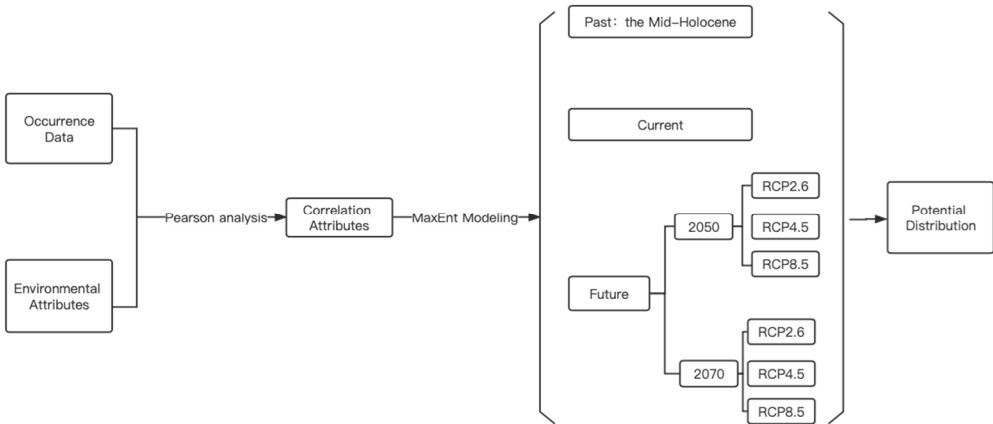

**Figure 3.** Research framework for geographical distribution of *Lagerstroemia excelsa* under climate change.

## 3. Results

### 3.1. Model Evaluation

Receiver Operating Characteristic (ROC) curve analysis of the test results (Figure 4) indicated that the Area Under the Curve (AUC) values for the Training data and Test data were 0.961 and 0.961, respectively. A high AUC value is indicative of the model's robust discriminative power and accuracy in distinguishing between species presence and absence based on environmental variables. A high AUC, close to 1, signifies accurate predictions with low rates of false positives and false negatives. These values, nearing 1, suggest that the model demonstrated high stability and accuracy. The precision of the simulation results established its suitability for research on purple wisteria segmentation.

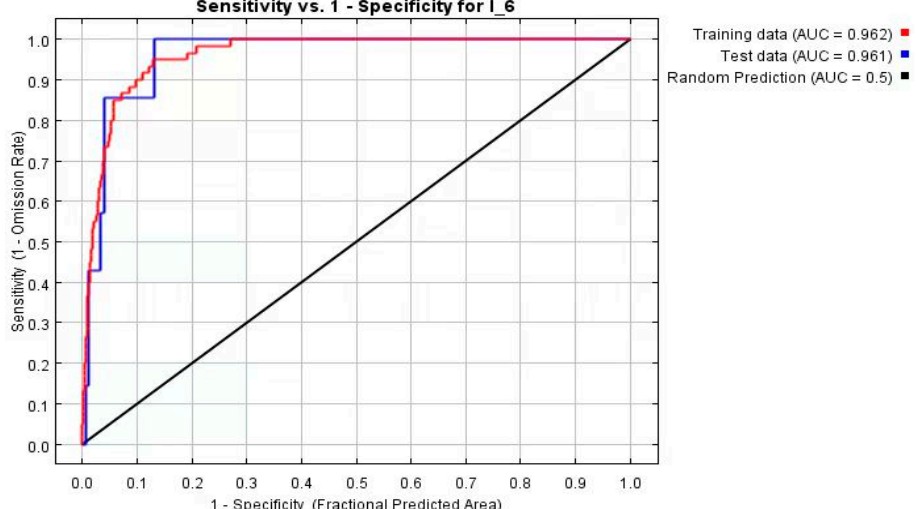

**Figure 4.** Evaluation of the ROC curve of the simulation results of suitable areas for *Lagerstremia excelsa*.

### 3.2. Attribute Contribution

Drawing upon prior research [29], the selection of 0.8 as the threshold for variable selection achieves a nuanced equilibrium between sensitivity and specificity. This judicious choice exhibits ample stringency to alleviate the influence of highly correlated variables without undue strictness, thereby enabling the inclusion of variables with substantive significance. Based on Pearson analysis, eight factors ($|r| < 0.8$) were selected, including annual mean temperature (bio1), mean diurnal range (bio2), temperature seasonality (bio4), annual precipitation (bio12), precipitation of the wettest month (bio13), precipitation of the warmest quarter (bio18), precipitation of the coldest quarter (bio19), and elevation (Ele) (Figure 5).

The "knife-cutting test" holds significance in ecological modeling as it aids researchers in assessing the importance of individual variables and identifying dominant factors that strongly influence model accuracy. This test not only reveals the impact of changes in specific variables on the model's output but also contributes to understanding the model's responsiveness. The highest contribution rate was attributed to Annual precipitation (bio12) at 38.3%. Following closely were Precipitation of the coldest quarter (bio19, 28.7%), Mean diurnal range (bio2, 13.5%), Temperature seasonality (bio4, 8.6%), Elevation (Ele, 5.4%), Annual mean temperature (bio1, 4.9%), Precipitation of the wettest month (bio13, 0.4%), and Precipitation of the warmest quarter (bio18, 0.1%). Table 2 and Figure 6 present the contribution rates of various environmental variables along with the results of the knife-cutting test. The cumulative contribution rates of the first six bioclimatic variables accounted for 99.5%, underscoring their pivotal role as primary environmental factors in model construction. Annual precipitation (bio12) and mean diurnal range (bio2) achieved a standard training gain ratio of 1.5, indicating that these two variables, when used individually, can capture more relevant climatic information compared to other environmental variables.

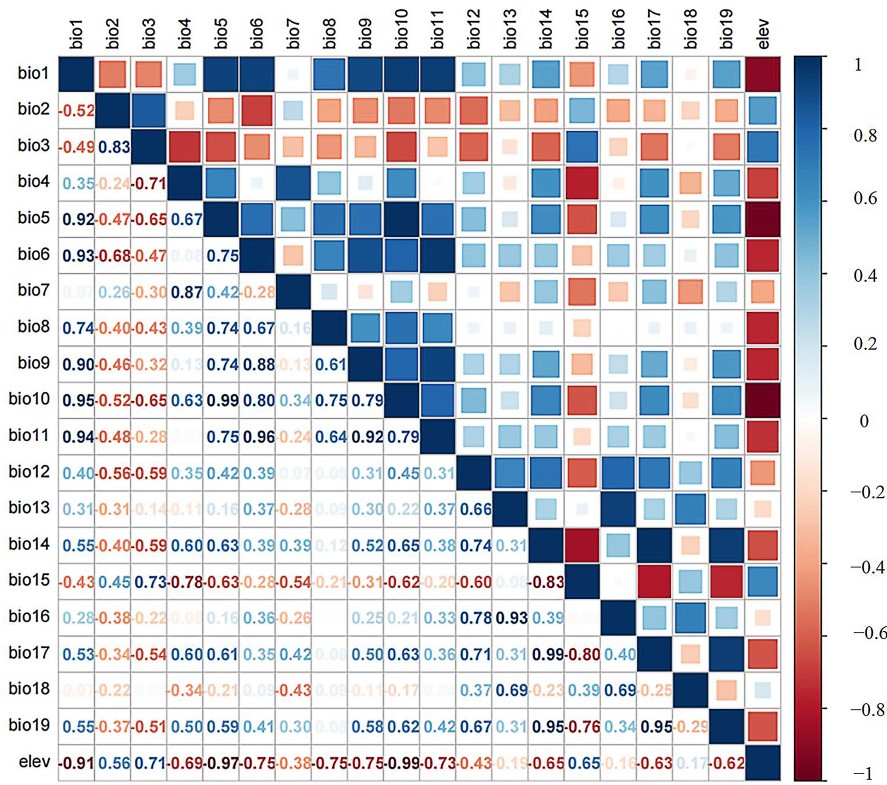

**Figure 5.** The results of Pearson analysis for environmental attribute contribution of *Lagerstroemia excelsa* under climate change.

**Table 2.** Contribution rates of each environmental variable to the distribution of *Lagerstroemia excelsa*.

| Attributes | Summary | Percent Contribution | Permutation Importance |
|---|---|---|---|
| bio1 | Annual mean temperature | 4.9 | 30.1 |
| bio2 | Mean diurnal range | 13.5 | 6.7 |
| bio4 | Temperature seasonality | 8.6 | 3.9 |
| bio12 | Annual precipitation | 38.3 | 47 |
| bio13 | Precipitation of wettest month | 0.4 | 0.4 |
| bio18 | Precipitation of warmest quarter | 0.1 | 1.5 |
| bio19 | Precipitation of coldest quarter | 28.7 | 0.1 |
| Ele | Elevation | 4.9 | 30.1 |

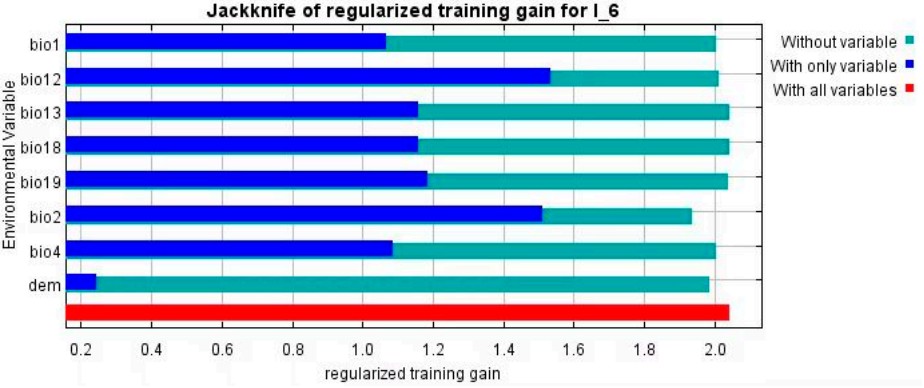

**Figure 6.** The Jackknife of environmental attributes of *Lagerstroemia excelsa* under climate change.

### 3.3. Potential Distributions

The MaxEnt model's predicted results are presented in Table 3, Figures 7–9. In general, the suitable habitat of *L. excelsa* remains relatively stable across the Mid-Holocene (MH), current period, and the years 2050 and 2070. Its changes are not very significant. In the current period, *L. excelsa* is primarily distributed in core areas encompassing Hunan, Hubei, Guizhou, Chongqing, and Sichuan. The total suitable habitat area is estimated to be $1.41 \times 10^6$ km$^2$, with a high-suitability zone covering $3.34 \times 10^5$ km$^2$, constituting 23.83% of the total suitable habitat area. During the Mid-Holocene, the total suitable habitat area was approximately $1.30 \times 10^6$ km$^2$, with a high-suitability zone covering $3.23 \times 10^5$ km$^2$, making up 24.83% of the total suitable habitat area.

**Table 3.** Suitable area of *Lagerstroemia excelsa* under different climate period.

| | | High-Suitability Area (km²) | Proportion of High-Suitable Area (%) | Medium-Suitability Area (km²) | Low-Suitability Area (km²) | Unsuitable Area (km²) | Suitability Area (km²) |
|---|---|---|---|---|---|---|---|
| Mid-Holocene (MH) | | $3.23 \times 10^5$ | 24.84 | $3.79 \times 10^5$ | $5.99 \times 10^5$ | $8.29 \times 10^6$ | $1.30 \times 10^6$ |
| Current | | $3.35 \times 10^5$ | 23.83 | $4.34 \times 10^5$ | $6.36 \times 10^5$ | $8.19 \times 10^6$ | $1.41 \times 10^6$ |
| 2050 | RCP2.6 | $3.47 \times 10^5$ | 23.55 | $3.74 \times 10^5$ | $7.54 \times 10^5$ | $8.12 \times 10^6$ | $1.48 \times 10^6$ |
| | RCP4.5 | $3.43 \times 10^5$ | 22.89 | $4.54 \times 10^5$ | $7.01 \times 10^5$ | $8.09 \times 10^6$ | $1.50 \times 10^6$ |
| | RCP8.5 | $3.47 \times 10^5$ | 23.12 | $4.29 \times 10^5$ | $7.24 \times 10^5$ | $8.09 \times 10^6$ | $1.50 \times 10^6$ |
| 2070 | RCP2.6 | $3.29 \times 10^5$ | 22.34 | $4.44 \times 10^5$ | $6.99 \times 10^5$ | $8.12 \times 10^6$ | $1.47 \times 10^6$ |
| | RCP4.5 | $3.48 \times 10^5$ | 23.88 | $4.31 \times 10^5$ | $6.78 \times 10^5$ | $8.14 \times 10^6$ | $1.46 \times 10^6$ |
| | RCP8.5 | $3.26 \times 10^5$ | 23.99 | $3.46 \times 10^5$ | $6.86 \times 10^5$ | $8.23 \times 10^6$ | $1.36 \times 10^6$ |

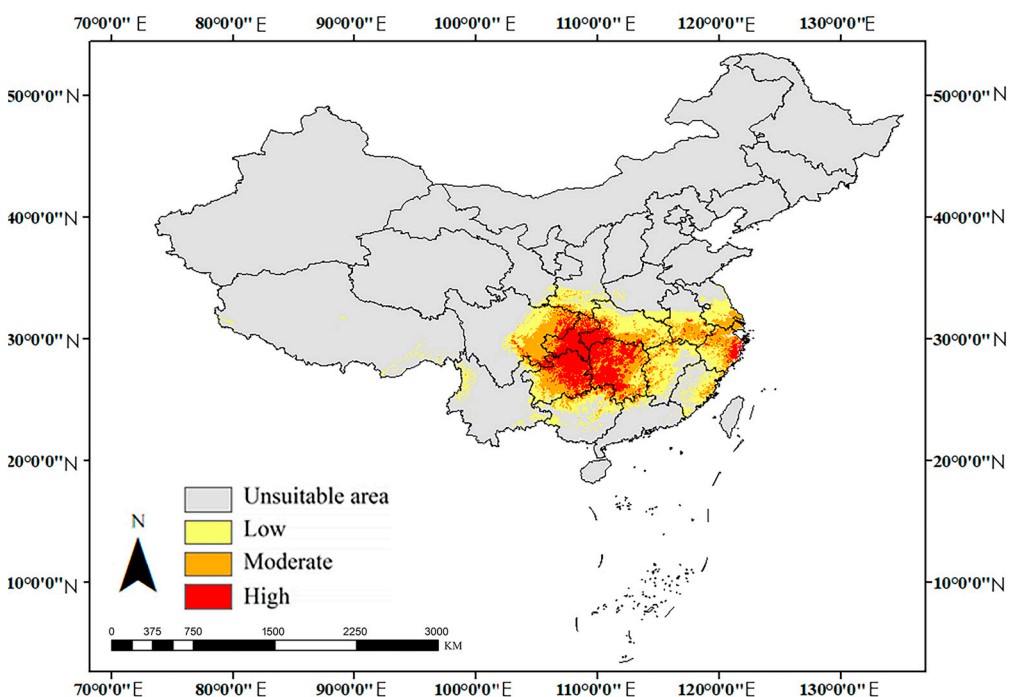

**Figure 7.** The suitable habitat of *Lagerstroemia excelsa* under current climate conditions.

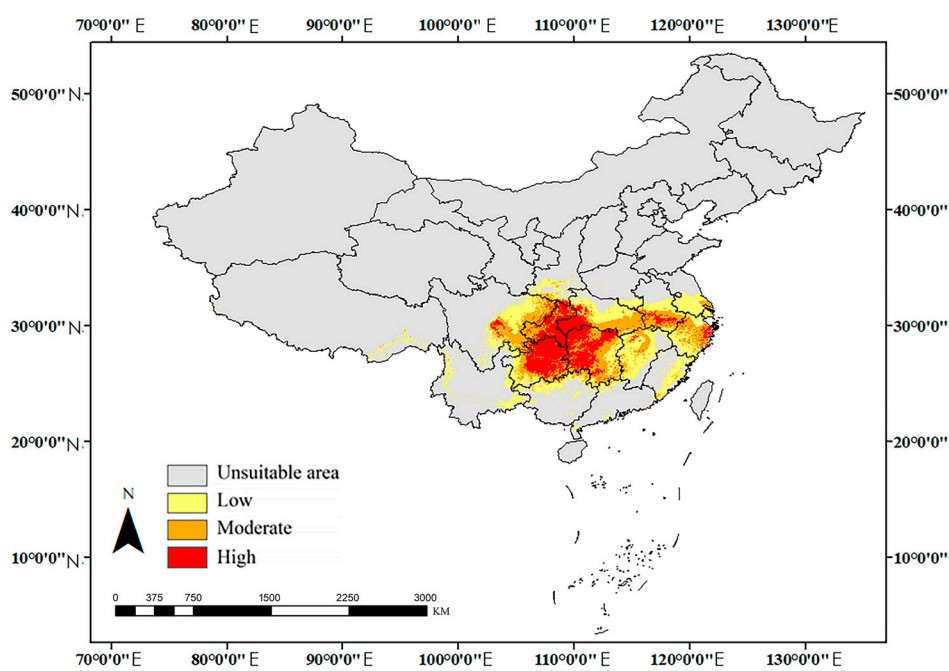

**Figure 8.** The suitable habitat of *Lagerstroemia excelsa* under Mid-Holocene climate conditions.

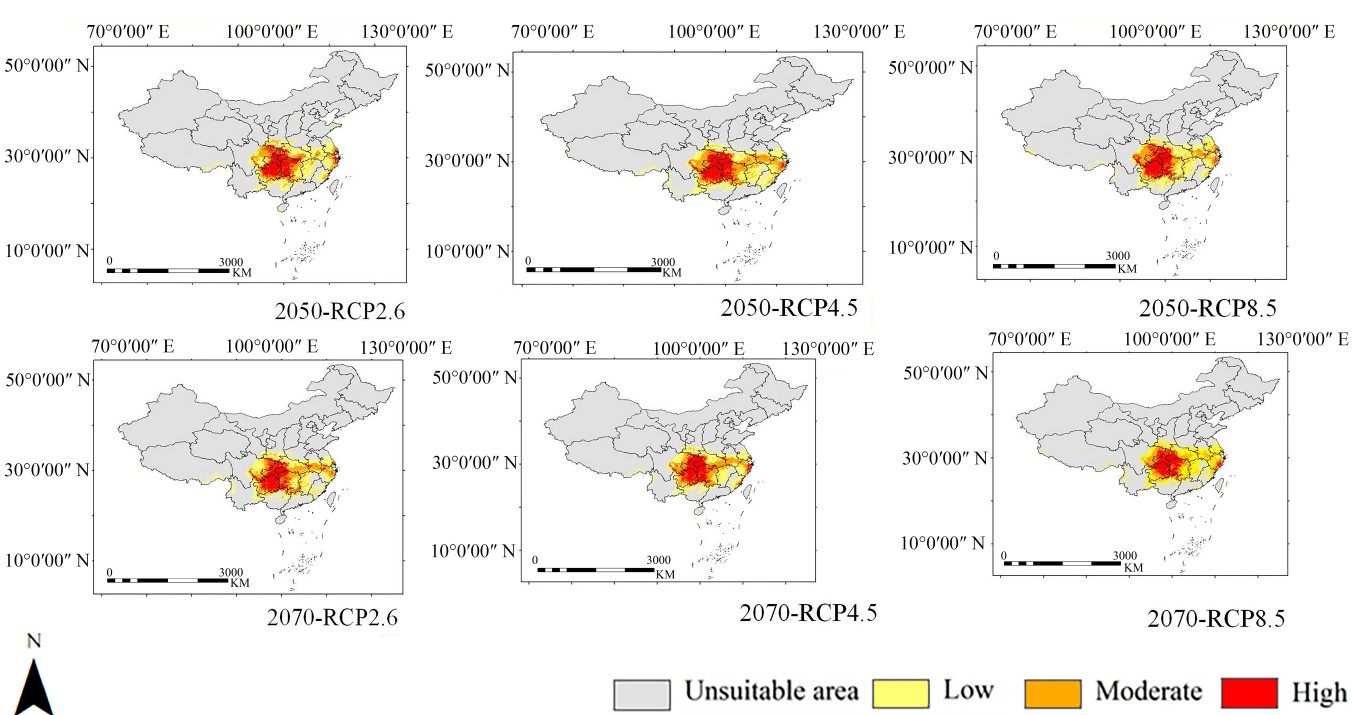

**Figure 9.** The suitable habitat of *Lagerstroemia excelsa* under future climate conditions.

The selection of RCPs was made to emphasize the developmental trajectories of future greenhouse gas emissions. RCP2.6, RCP4.5, and RCP8.5 were specifically chosen to represent low emissions, moderate greenhouse gas concentration increases, and higher greenhouse gas concentration scenarios, respectively.

In the year 2050, under the future 2050-RCP2.6 scenario, the total suitable habitat area is projected to increase to $1.48 \times 10^6$ km$^2$, with a high-suitability zone covering $3.48 \times 10^5$ km$^2$, accounting for 23.56% of the total suitable habitat area. Under the future 2050-RCP4.5 scenario, the total suitable habitat area will be $1.50 \times 10^6$ km$^2$, with a high-suitability zone covering $3.43 \times 10^5$ km$^2$, constituting 22.90% of the total suitable habitat

area. Likewise, under the future 2050-RCP8.5 scenario, the total suitable habitat area is expected to reach $1.50 \times 10^6$ km$^2$, with a high-suitability zone covering $3.47 \times 10^5$ km$^2$, representing 23.12% of the total suitable habitat area.

In the year 2070, under the future 2070-RCP2.6 scenario, the total suitable habitat area is forecasted to be $1.47 \times 10^6$ km$^2$, with a high-suitability zone covering $3.29 \times 10^5$ km$^2$, accounting for 23.12% of the total suitable habitat area. Under the future 2070-RCP4.5 scenario, the total suitable habitat area will be $1.50 \times 10^6$ km$^2$, with a high-suitability zone covering $3.43 \times 10^5$ km$^2$, constituting 22.90% of the total suitable habitat area. Under the future 2070-RCP8.5 scenario, the total suitable habitat area is projected to be $1.36 \times 10^6$ km$^2$, with a high-suitability zone covering $3.26 \times 10^5$ km$^2$, making up 23.99% of the total suitable habitat area.

## 4. Discussion

### 4.1. Environmental Attributes Influencing the Distribution of Lagerstroemia excelsa

Climate change presents a significant challenge to ecosystems. Understanding the alteration of plants' suitable habitats is crucial for the protection and utilization of ecosystems. To address this challenge, the MaxEnt model was employed to predict the suitable habitat of the representative tree species, *L. excelsa*. The key environmental factors influencing its distribution were identified using the knife-edge method. The knife-cutting test showed that temperature and precipitation are the primary environmental factors influencing *L. excelsa* distribution. Annual precipitation, elevation, and annual mean temperature were the most significant environmental factors influencing the suitable area of *L. excelsa*. The research results align with previous studies on *L. excelsa*, providing further evidence of the reliability of the predictive outcomes [12]. The research findings highlight that the key environmental variables influencing the distribution of *L. excelsa* were annual precipitation (bio12), precipitation of the coldest quarter (bio19), mean diurnal range (bio2), temperature seasonality (bio4), elevation (Ele), and annual mean temperature (bio1), collectively contributing to 99.5% of the model's explanatory power. This underscored the predominant influence of precipitation and temperature on *L. excelsa* distribution. Notably, annual precipitation (bio12) emerged as the most substantial contributor with a standard training-gained ratio of 1.5, emphasizing its pivotal role in determining *L. excelsa*'s distribution in the Sichuan–Guizhou region. Furthermore, elevation (Ele) exhibited a permutation importance of 30.1%, comparable to that of annual precipitation (bio12), indicating the crucial role of elevation in influencing *L. excelsa*'s distribution. This may be due to the inclusion of altitude as a descriptor or independent variable in the bioclimatic variables. This aligned with previous findings, reinforcing the reliability of the research outcomes [12].

The distribution of plants is intricately shaped by factors such as temperature, altitude, and precipitation, as these environmental elements directly or indirectly impact the adaptability and growth conditions of plants [30]. Temperature, a crucial factor, influences key life processes of plants, including growth rate [31], photosynthetic efficiency, seed germination, flowering, and fruiting, consequently shaping their suitable habitats. Generally, an increase in temperatures correlates with a decrease in the number and diversity of plants in high-altitude regions, while low-altitude regions tend to experience an increase in both [32]. As altitude rises, temperature and atmospheric pressure decrease, leading to variations in humidity and sunlight duration. High-altitude regions, being colder, often pose limitations on plant growth. Some plant species exhibit better adaptability to high-altitude environments, while others thrive in low-altitude regions. Water, a fundamental requirement for plant growth and survival, is directly influenced by precipitation. Precipitation significantly impacts the water supply to plants, thereby influencing their growth, life cycle, and distribution. Generally, regions with high precipitation support lush vegetation [33], while arid regions may only sustain specific plant species adapted to dry conditions.

According to the research findings, annual precipitation, elevation, and annual mean temperature were identified as the most significant environmental factors influencing the suitable area of *L. excelsa*. Therefore, these three environmental factors can be considered

primary considerations for future introduction and domestication efforts of *L. excelsa*. Despite *L. excelsa*'s high ecological value and potential applications, its resources are currently mainly derived from natural populations and it has not been widely cultivated or utilized in urban landscapes. In the future, it is advisable to focus on selecting regions with similarities in annual precipitation, elevation, and annual mean temperature to the current distribution areas of *L. excelsa* for artificial cultivation and landscape applications.

### 4.2. Response of Lagerstroemia excelsa Distribution to Climate Change

The MaxEnt model suggested minimal variation in the potential distributions of *L. excelsa* across different climate scenarios. However, there was a slight increase projected in the suitable area for *L. excelsa* under future climate scenarios. The research findings reveal that the current suitable area for *L. excelsa* is primarily concentrated in the middle and lower reaches of the Yangtze River, with the Wuling Mountains serving as its distribution center. Projections suggest a slight increase in the suitable area under most future climate scenarios. Similar observations were noted in the case of the *Taxus* genus, a coexisting species with *L. excelsa*, where precipitation was also identified as a crucial factor [14].

Annual precipitation (bio12) emerged as the most significant factor influencing the suitable area of *L. excelsa* under all climate scenarios except for 2050-RPC 2.6. The precipitation level was notably greater than in the current climate scenario, potentially serving as the primary reason for the observed slight increase in the suitable area.

In the year 2070, the high suitability zones for *L. excelsa* experienced a slight decrease under the climate scenarios of RCP2.6 and RCP8.5. Therefore, for the natural populations of *L. excelsa* situated in high-suitability zones, efforts should be made to enhance their in situ protection, aiming to mitigate the potential impacts of future climate change. Additionally, noticeable changes in the high-suitability zones were observed in the downstream area of the Yangtze River under these two climate scenarios. To the best of our knowledge, there are no natural populations of *L. excelsa* distributed in this region. Conservation efforts should be focused on the existing plantations, nurseries, and cultivated *L. excelsa* in this area. Future breeding and landscape utilization of *L. excelsa* should also take this into consideration, avoiding the potential impact of climate change on the utilization of *L. excelsa* in these regions.

### 4.3. Conservation and Utilization of Lagerstroemia excelsa

*Lagerstroemia excelsa* currently relies on limited resources, primarily inhabiting natural forests, while artificial populations are notably smaller in comparison. Natural forest populations, hosting more private alleles, demonstrate higher genetic diversity, making them optimal for research and conservation purposes [34]. Although the suitable area for *L. excelsa* is anticipated to experience a slight increase under most future climate scenarios, excluding the 2070-RCP 4.5 and 2070-RCP 8.5 climate scenarios, the proportion of high-suitability areas exhibits a declining trend. Therefore, the conservation of natural populations of *L. excelsa* is deemed crucial and necessary. Research findings highlight that high-suitability areas for *L. excelsa* were primarily concentrated in the provinces of Hunan, Hubei, Guizhou, and Chongqing, situated on both sides of the Wuling Mountains. Previous studies and field surveys had identified these regions as core areas for the distribution of natural populations and ancient trees of *L. excelsa*. Consequently, we recommend reinforcing in situ conservation efforts and establishing natural reserves for ancient trees or natural populations of *L. excelsa* in these regions. Especially in the high-suitability areas of the Wuling Mountains region, including Fanjing Mountain in Guizhou, Hupingshan Mountain, Tianmen Mountain, Gaowangjie, Badagong Mountain in Hunan, and Yesan River Basin in Hubei. The goal is to minimize human-induced damage and disturbances. Given the considerable ornamental value of *L. excelsa* and its popularity among bonsai enthusiasts, it is essential to intensify cultural promotion efforts, raising public awareness about its conservation. This proactive approach aims to prevent potential illegal logging and excessive felling of the species. Furthermore, during translocation conservation and

cultivation processes, it is crucial to adhere to the principle of matching trees to suitable sites. Comprehensive consideration of factors such as disease and pest resistance, inter-specific competition, and soil texture is vital to enhance the survival rate of *L. excelsa* cultivation.

In the current period, *L. excelsa* exhibits a total suitable habitat area of $1.41 \times 10^6$ km$^2$, with a high-suitability zone covering $3.35 \times 10^5$ km$^2$. This underscores the species' potential for diverse landscape applications, bridging the application gap for large deciduous *Lagerstroemia* species in the middle and lower reaches of the Yangtze River region in China. Notably, *L. excelsa* can play a significant role in addressing this gap. Existing research and resource surveys indicated a scarcity of artificially cultivated populations of *L. excelsa*, with Chengbu County in Hunan Province, China, being one of the few exceptions. As a result, targeted conservation and utilization efforts can be directed toward *L. excelsa* based on the predictions from the MaxEnt model. Establishing an optimal growth environment will facilitate the sustainable development and utilization of its resources. Moreover, introducing *L. excelsa* to moderately suitable areas can contribute to population reproduction and expansion, promoting the species' overall resilience and adaptability.

## 5. Conclusions

Based on the existing occurrences of *L. excelsa*, this study employed Pearson analysis and the Maximum Entropy Model (MaxEnt) to anticipate the species' suitable habitat across eight distinct climate scenarios spanning the Mid-Holocene (MH), the current period, and the years 2050 and 2070. Additionally, the research conducted an in-depth analysis of the primary environmental factors influencing the distribution of *L. excelsa*.

In the case of *L. excelsa*, both precipitation and temperature emerged as the predominant constraining elements for its geographic spread. The anticipated effects of climate change suggested a slight expansion of the suitable distribution area, predominantly concentrated around the Wuling Mountains, although the overall shift remains modest. While the anticipated suitable area for *L. excelsa* was expected to marginally increase under the majority of future climate scenarios, it is predicted to diminish specifically under the 2070-RCP 4.5 and 2070-RCP 8.5 scenarios. Consequently, the preservation of natural populations of *L. excelsa* becomes not only crucial but also warrants immediate attention. The findings of this study furnish valuable evidence to bolster forthcoming endeavors, encompassing the introduction, selective breeding, and systematic cultivation of *L. excelsa*, all while accounting for the ramifications of global warming.

**Author Contributions:** S.H., Conceptualization, Data curation, Methodology, Software, Visualization, Project administration, Writing—original draft, and Writing—review and editing. D.Z., Conceptualization, Resources, Writing—review and editing, and Supervision. Y.W., Conceptualization, Resources, Writing—review and editing, and Supervision. All authors have read and agreed to the published version of the manuscript.

**Funding:** This research received no external funding.

**Data Availability Statement:** The data presented in this study are available on request from the corresponding author.

**Conflicts of Interest:** The authors declare no conflict of interest.

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
