# Peer review of "Potential Geographical Distribution of Lagerstroemia excelsa under Climate Change"

_agriculture, doi:10.3390/agriculture14020191_

Round 1

Reviewer 1 Report

Comments and Suggestions for Authors

Line 42.- It is important for the reader to see a photograph or figure of the whole tree and a close-up of its leaves and flowers, like to be following example (It's just an example, but the authors decided to use a specific figure).

Line 58.- I do not fully agree with this sentence, I think the word "highly" is very inappropriate. I agree that ENM is an adequate method to approach the knowledge of the geographical niche of a species, however, the article only uses MAXENT, a single model. I think it is better to remove the word "highly".

Lines 69-72: What is the reason there is no question about the spatial distribution of the niche in the mid-Holocene?.

Lines 74-85: The selection of L. excelsa occurrence points were made according to the literature. It is possible that 71 occurrence points may be considered insufficient, however, for the study area it seems to be appropriate.

Lines 89-105: The database of gridded layers corresponding to the environmental variables used, including altitude, is in accordance with the scientific literature.

What is the reason for knowing the niche in the mid-Holocene?

Line 110: Please explain in more detail what Pearson's analysis consists of. I think that you, by means of Pearson's correlation test, selected the least correlated environmental variables (r >0.6, or 0.70, or 0.8,.......), which were included in the model.

Line 114: What is the technical reason for using a threshold value of 0.8 for the Pearson correlation coefficient in the selection of environmental variables?

Lines 203-206: The fact that you mention could be because the bioclimatic variables contain altitude as a descriptor or independent variable. What is your opinion?

Lines 206-207: Could you provide some quotes to support this statement?

Lines 267-271: It seems that this paragraph does not contribute to the discussion located here, I think that this issue should be raised at the beginning of the discussion.

Line 278: This modelling does not contribute anything to the current structure of the article. I would seriously consider removing it from the whole article.

Reviewer 2 Report

Comments and Suggestions for Authors

Dear Author:

I have completed the review of your manuscript titled "Potential Geographical Distribution of Lagerstroemia excelsa under Climate Change" submitted to agriculture journal.

I want to commend you on the significance of your study in addressing the crucial issue of Lagerstroemia excelsa's potential distribution under changing climatic conditions. Your research has the potential to contribute significantly to our understanding of the natural resources and growing conditions for Lagerstroemia excelsa, offering valuable insights for protection and sustainable utilization.

However, I must highlight a few weaknesses identified during the review process. By addressing these points, you can enhance the clarity, cohesion, and overall impact of the introduction.

-          Please refrain from utilizing the keyword present in the title.

-          You have repeated the name of the species "Lagerstroemia excelsa" several times in the abstract. Kindly substitute it with "L. excelsa" or employ terms like the current species or the studied species.

-          On line 13, avoid using personal language such as "we employed...." Instead, use third-person language.

-          Ensure that the subtitle of the introduction is included in the corresponding section.

-          In lines 46 and 47, please provide a citation for the mentioned sentence.

-          Could you elaborate on the novelty of this study?

-          I recommend incorporating the following reference for lines 57-59:

Machine learning models in the prediction of the ecological niche for wild pistachio. Published in "Ecological Informatics," Volume 72, 101907.

-          This study focuses on the niche modeling of Lagerstroemia excelsa. In the introduction, please provide information about the specific habitat where this species thrives.

-          The introduction provides a solid foundation for the study, highlighting the broader context of climate change and its impact on ecosystems. However, a few areas could be strengthened or clarified:

-          Some sentences are lengthy and may benefit from breaking them down for clarity. For example, consider dividing sentences in lines 29-31 and lines 44-46 for easier comprehension.

-          The transition between the general discussion on climate change and the specific focus on Lagerstroemia excelsa could be smoother. Consider adding a brief sentence to explicitly introduce the transition from the broader context to the study species.

-          While the ecological and cultural significance of Lagerstroemia excelsa is well-described, the rationale for selecting this species for the study could be emphasized more. Why is L. excelsa particularly suited for studying the impacts of climate change, and what unique characteristics or challenges does it present?

-          The introduction would benefit from a more comprehensive literature review, especially regarding previous studies on Lagerstroemia excelsa and its response to climate change. This will help contextualize the study within the existing body of knowledge.

-          While the study objectives are stated at the end, they could be more clearly delineated. Consider rephrasing or restructuring the sentences in lines 67-71 to explicitly outline the research questions and objectives.

-          Consider adding a sentence or two to engage the reader further, perhaps by emphasizing the significance of understanding the potential distribution of L. excelsa in the context of broader environmental concerns.

-          Consider breaking down the long sentences in lines 79-82 and lines 89-97 for better readability.

-           Use subheadings or bullet points to clearly delineate different stages of the methodology, such as "Species Occurrence Data," "Environmental Attributes," and "MaxEnt Modeling."

-           Explicitly state the criteria for selecting and calibrating coordinates using Google Earth (lines 79-82) for transparency and reproducibility.

-           Clarify how field investigations were conducted to supplement the dataset (line 77).

-           Briefly explain the rationale for selecting the specific environmental variables (bio1-bio19, Elevation) and terrain variable (lines 90-91).

-          Provide more context on why Mid-Holocene (MH) data from the MIROC model were used (lines 92-96).

-           Explain the reason behind choosing the BBC_CSM model and the three concentration path scenarios (lines 95-98).

-          Include a brief explanation of why these scenarios were considered representative of future climate conditions.

-          Offer more details on how Pearson correlation analysis was applied, and the threshold used for identifying primary environmental factors (lines 102-104).

-          Clarify the significance of iterating the operational process ten times (line 110).

-          Provide more details on how the training and verification datasets were split (lines 110-111).

-            Mention any specific version numbers for R and ArcGIS used to enhance transparency and reproducibility.

-           While the AUC values are presented for the ROC analysis, provide a brief explanation for readers unfamiliar with this metric. Discuss the significance of the high AUC values and how they reflect the model's accuracy (lines 134-138).

-           Clarify the significance of the knife-cutting test and how it contributes to understanding the importance of different environmental variables (lines 151-157).

-           Offer a concise summary of the results in text format before referring to Table 2 and Figure 5.

-          Instead of providing numerical results first, consider introducing the general findings and trends of Lagerstroemia excelsa's potential distribution under different scenarios before delving into specific numbers (lines 167-182).

-          Provide a clear rationale for the selection of the different future climate scenarios and their respective concentration paths (lines 174-179, 182-185).

-          - Consider a more gradual transition between the results section and the discussion section.

-          - Clearly state the key findings before delving into the discussion of environmental attributes and their implications for Lagerstroemia excelsa distribution (lines 194-206, 225-228).

-          - In the discussion of environmental attributes influencing distribution, connect the findings to the broader literature and the specific characteristics of Lagerstroemia excelsa (lines 195-206).

-          - Emphasize the ecological significance of the identified key environmental variables in the context of L. excelsa's adaptability and growth conditions.

-          Connect the discussion of conservation recommendations more explicitly to the findings, such as the identified high-suitability areas and their geographical locations (lines 236-254).

-          Discuss the implications of the declining trend in high-suitability areas under certain future scenarios for conservation efforts.

Comments on the Quality of English Language

The language is fine; only minor edits are needed, especially considering breaking down the long sentences for better readability.

Round 2

Reviewer 2 Report

Comments and Suggestions for Authors

Dear Authors,

I have reviewed the revised version of our manuscript and am generally satisfied with the improvements made. However, I would like to suggest some enhancements to further improve the overall clarity and reader experience. Specifically, I recommend ensuring that all figures exhibit increased resolution, and the font size for digits is enlarged to enhance visibility. Additionally, I propose that comprehensive and standalone captions be added to all figures and tables. This adjustment will enable readers to understand the content independently, without relying on the accompanying text.

Good luck
